# Up-Frameshift Factors from Phytopathogenic Fungi Play a Crucial Role in Nonsense-Mediated mRNA Decay

**DOI:** 10.3390/jof11060404

**Published:** 2025-05-23

**Authors:** Ping Lu, Jiaqi Wang, Xiaoli Wang, Dan Wang, Haojie Shi

**Affiliations:** 1The Key Lab for Biology of Crop Pathogens and Insect Pests and Their Ecological Regulation of Zhejiang Province, College of Advanced Agricultural Sciences, Zhejiang A&F University, Hangzhou 311300, China; luping@zafu.edu.cn (P.L.); wjq20001002@163.com (J.W.); 2Jiangsu Provincial Key Construction Laboratory of Probiotics Preparation, Huaiyin Institute of Technology, Huaian 223003, China; xlwang@hyit.edu.cn; 3National Key Laboratory for Development and Utilization of Forest Food Resources, Zhejiang A&F University, Hangzhou 311300, China

**Keywords:** nonsense-mediated mRNA Decay (NMD), phytopathogenic fungi, *UPF* genes, exon junction complex (EJC)

## Abstract

The nonsense-mediated mRNA decay (NMD) is extensively involved in physiological, pathological, and stress response processes in humans and plants. However, the NMD in phytopathogenic fungi has not yet been thoroughly investigated. In this study, we identified and performed domain analysis on the core components of the NMD in ten globally widespread phytopathogenic fungi that cause significant economic losses. The core components of NMD in these fungi exhibit high similarity to their homologous genes in humans, while also possessing certain specificities. The core factors of the NMD, including the Up-frameshift factors (UPFs) and the exon junction complex (EJC), are generally conserved among phytopathogenic fungi. Notably, suppressors with morphological effects on genitalia (SMG) genes are absent in these fungi, which bears some similarity to the EJC-independent NMD degradation mechanism observed in *Saccharomyces cerevisiae*. Interestingly, plant pathogenic fungi contain highly homologous genes of the EJC complex, suggesting the presence of an EJC-dependent NMD degradation mechanism. In summary, our findings demonstrate that NMD are prevalent in plant pathogenic fungi, providing a research foundation for subsequent studies on NMD in their growth, development, and involvement in pathogenic processes.

## 1. Introduction

Eukaryotic cells utilize various mechanisms to regulate gene expression, including transcriptional regulation, post-transcriptional control of mRNA translation, mRNA degradation, and post-translational modifications of proteins [1]. Transcriptional regulation, a highly controlled process that enables the production of multiple proteins through distinct processing of the same gene in response to environmental changes, ensures gene expression occurs exclusively under suitable environmental conditions. *MeCP2*, a gene encoding the Methyl-CpG binding protein 2, associated with functions related to epigenetic silencing in human [2]. The first exon of the *MeCP2* gene can produce two transcripts through alternative splicing: MeCP2-E1 and MeCP2-E2. MeCP2-E1 is the predominantly expressed transcript in the brain, playing a crucial role in maintaining normal neuronal function, whereas MeCP2-E2 can interact with the transcription factor FoxG1 to inhibit the expression of neurotoxins [3,4,5]. In sunflowers, Glycosyl Hydrolase 17 may regulate seed size through different transcript isoforms, enabling adaptation to two distinct ecological environments: dune and non-dune [6].

However, not all proteins produced by transcriptional regulatory mechanisms have independent functions. In plants, about 40% of transcripts produce premature termination codon (PTC) [7]. Transcriptions containing PTC are translated into truncated proteins, which accumulate in organisms and can cause toxic effects [8]. Organisms have evolved sophisticated quality control mechanisms such as the extensively studied NMD pathway, which serves as a critical surveillance system by selectively identifying and degrading mRNAs containing PTC, thereby preventing the accumulation of potentially deleterious truncated proteins [9].

The NMD was first discovered in *Saccharomyces cerevisiae* and human in 1979 [10,11]. Subsequently, it has been studied in an increasing number of model species, including *Caenorhabditis elegans*, *Drosophilidae*, *Arabidopsis thaliana*, *Dictyostelium discoideum*, mice, zebrafish, etc. [12,13,14,15,16,17] (Table 1). The mRNA degradation mechanism mediated by the NMD pathway is highly conserved, and despite significant differences among species of varying evolutionary degrees, the functions involved in these organisms are critically important. The NMD plays a crucial role in regulating the development processes of animal neurons and is implicated in premature aging [18,19]. Additionally, this mechanism is involved in plant antiviral processes and responses to environmental changes [20,21]. However, in fungi, particularly plant pathogenic fungi, the NMD has not been thoroughly elucidated. This study describes the homologous genes of NMD core factors in ten significant plant pathogenic fungi and provides a detailed comparison of the functional domain conservation of the core up-frameshift factors (UPF1, UPF2, and UPF3) and EJC.

The EJC is a complex composed of multiple subunit proteins that assembles 20–24 nucleotides upstream of the exon-exon junction during the splicing process [22]. The EJC serves as a molecular link between splicing, mRNA export, cytoplasmic mRNA localization, and translation, thereby enhancing mRNA expression [23,24]. At present, the activation pathways of NMD has been found to be divided into two modes: EJC-dependent and EJC-independent models [25]. The EJC-independent activation mechanism has been identified in species with lower evolutionary complexity, such as yeast and *C. elegans* [26,27]. The current research results show that the two patterns are similar and complementary, and some core factors are indispensable, including UPFs, SMGs, etc [25]. Therefore, we aim to explore the identification of additional core factors related to NMD in ten important plant pathogenic fungi in this study, including *Magnaporthe oryzae*, *Botrytis cinerea*, *Puccinia triticina*, *Fusarium graminearum*, *Fusarium oxysporum*, *Blumeria graminis*, *Mycosphaerella graminicola*, *Colletotrichum higginsianum*, *Ustilago maydis*, *Verticillium dahliae*.

## 2. Materials and Methods

### 2.1. Acquisition of Protein Sequences

The protein sequences of the core components of human NMD were obtained from the NCBI database (https://www.ncbi.nlm.nih.gov/) (accessed on 2 March 2022). The protein sequences of plant pathogenic fungi are downloaded from Ensembl fungi (https://fungi.ensembl.org/) (accessed on 15 April 2022) and FgBase (http://fgbase.wheatscab.com/) (accessed on 15 April 2022) [28].

### 2.2. Identification of Homologous Genes and Protein Domains

Using the NMD sequence in humans as the reference sequence, a database was constructed through local blast v2.5.0. The proteins of plant pathogenic fungi were aligned with the human reference sequence using BLASTp. Genes are considered homologous if their E-value ≤ 1 × 10^−5^. The protein domains of the genes were identified using interproscan v68 [29].

### 2.3. Prediction of Protein Phosphorylation Sites

The phosphorylation sites of proteins were predicted using NetPhos-3.1b (https://services.healthtech.dtu.dk/services/NetPhos-3.1/) (accessed on 8 June 2022).

### 2.4. Prediction of Protein-Protein Interactions

The protein interaction probabilities among *UPF* genes of plant pathogenic fungi were predicted using the web version of AlphaFold3 (https://alphafoldserver.com/) (accessed on 3 February 2024), and the DockQ values were calculated. DockQ is a metric used to assess the quality of protein-protein docking models. It is defined as DockQ = (0.8 ipTM + 0.2 pTM), where ipTM represents the interface pTM score and pTM denotes the predicted Template Modeling score [30,31].

## 3. Results

### 3.1. Core Elements Involved in Nonsense-Mediated Decay (NMD) Are Conserved in Plant Pathogenic Fungi

In humans, the NMD consists of the UPF proteins including UPF1, UPF2, and UPF3. Additionally, it encompasses the suppressors that influence genital morphological characteristics, namely SMG1, SMG5, SMG6, SMG7, SMG8, and SMG9. Furthermore, the EJC is part of this system, made up of subunits such as eIF4A3, MAGOH, RBM8A (commonly known as Y14), and CASC3 (also called BTZ or MLN51) [32].

Among plant pathogenic fungi, most of the homologous genes of the EJC-dependent NMD core factors were found, and only the homologous genes of the individual core factors were not found (Figure 1). In humans, the SMG1 protein is responsible for interacting with UPF1 and facilitating its phosphorylation [33]. However, *SMG1* has not been identified in plant pathogenic fungi (Figure 1). In addition to *SMG8*, *SMG9*, and *CASC3*, various genes exhibiting differing degrees of homologous similarity to factors involved in the NMD are present in plant pathogenic fungi. Among them, the *UPF1* and the *eIF4A3*, *MAGOH*, and *RBM8A* in the EJC have the highest homology (Figure 1; Appendix A). Additionally, certain individual homologous genes specific to some plant pathogenic fungi have not been identified, such as *UPF3* in *M. graminicola*. Our results indicate that the composition of the NMD core components in plant pathogenic fungi closely resembles that found in humans, including homologous genes of the EJC, such as *IF4A3*, *MAGOH*, and *RBM8A*. Currently, aside from the documented presence of the EJC in fission yeast, there have been no reports of its existence in other fungal species [34].

### 3.2. UPF1 in Plant Pathogenic Fungi

UPF1 is highly conserved across eukaryotes and serves as a crucial component in the NMD. It consists of three major domains: the cysteine-histidine-rich domain (CH), the helicase domain, and the serine-glutamine-rich domain (SQ) [35,36,37,38]. The CH domain is essential for its interaction with UPF2, whereas the SQ domain undergoes phosphorylation by SMG1 following the interaction with UPF1 [35,36,39].

In plant pathogenic fungi, the CH domain is highly conserved in UPF1, including in *M. graminicola*, which exhibits a lower degree of homology (Figure 2A). Judging from the conservation of the CH domain, UPF1 retains the potential to interact with UPF2 in plant pathogenic fungi. In comparison to the UPF1 in humans, the SQ domain is absent in plant pathogenic fungi (Figure 2A,B). In plant pathogenic fungi, the lack of an SQ domain implies that UPF1 functions differently in NMD compared to humans (Figure 2A). The phosphorylation of the SQ domain in UPF1 is a critical step in initiating the NMD in human [39]. Phosphorylation site prediction has revealed the presence of various phosphorylation sites at the termini of UPF1 proteins in all plant pathogenic fungi. This suggests that while SQ phosphorylation sites, which are present in humans, are absent in these fungi, alternative types of phosphorylation sites are nonetheless present. Consequently, phosphorylation in the UPF1 of plant pathogenic fungi may occur in forms distinct from the SQ domain. Our results indicate that *UPF1* are present in ten plant pathogenic fungi and retain the domain that interacts with UPF2. However, the *UPF1* in plant pathogenic fungi differs from that in humans, as it lacks the SQ domain. Additionally, the absence of *SMG1* in plant pathogenic fungi suggests that the potential for UPF1 protein to undergo phosphorylation or other modifications in these organisms remains unknown.

### 3.3. UPF2 in Plant Pathogenic Fungi

UPF2 functions as a bridging protein between UPF1 and UPF3 in the NMD [36,40]. In humans, the UPF2 protein has a length of 1272 amino acids and contains three middle portion of eukaryotic initiation factor 4G domains (MIF4G) domains and an glutamate-rich region (U1BD) [37]. In plant pathogenic fungi, the MIF4G and U1BD domains responsible for interacting with UPF1 and UPF3 are well conserved (Figure 3). It is noteworthy that the MIF4G1 located at the forefront is not universally retained in plant pathogenic fungi, except for *F. graminearum*, *F. oxysporum*, and *M. graminicola* (Figure 3).

The MIF4G1 and MIF4G2 of UPF2 support the UPF-EJC complex, but there is no evidence of direct interaction with the SURF complex [41]. The absence of MIF4G1 in plant pathogenic fungi indicates significant variations in the UPF-EJC complex configuration, though UPF2 retains its functional role as a bridging protein despite this deficiency. MIF4G-3 plays the most crucial role among the three MIF4Gs, as it is responsible for interacting with UPF3 [40,42]. Among the 10 plant pathogenic fungi, MIF4G-3 is highly conserved, indicating a high likelihood of interaction between UPF2 and UPF3 in plant pathogenic fungi. Similarly, the U1BD domain in UPF2, which interacts with UPF1, has been conserved in plant pathogenic fungi. These results suggest that, despite some variations in specific domains, the role of UPF2 as a connector between UPF1 and UPF3 in plant pathogenic fungi is analogous to its function in humans. Above all, compared to UPF2 in humans, the key structural domains of UPF2 are entirely identical, making it the UPF gene with the least variation in functional regions among the three.

### 3.4. UPF3 in Plant Pathogenic Fungi

UPF3 is an auxiliary component of the EJC and serves as an enhancer of the NMD [42,43]. In humans, the two distinct homologous genes of *UPF3* (*UPF3A* and *UPF3B*) originated from a duplication event of an ancestral gene and exhibit functional differences [44]. Although UPF3A and UPF3B share similar domains, UPF3B functions as a core NMD factor in human. The UPF2-UPF3B complex undergoes structural rearrangement upon binding with UPF1, thereby activating the ATP helicase activity of UPF1 [35,45]. Additionally, the UPF2-UPF3B complex can stimulate the phosphorylation of UPF1 by SMG1, subsequently triggering the NMD [46,47]. While UPF3 is conserved among plant pathogenic fungi, not all such fungi possess a homologous gene for UPF3 (Figure 4). Unlike in humans where *UPF3* gene duplication has produced two functional homologs (UPF3A and UPF3B), plant pathogenic fungi maintain only a single copy of UPF3 without evidence of duplication events.

The UPF3B N-terminus harbors a conserved RRM-L domain that interacts weakly with RNA, owing to the lack of aromatic residues required for high-affinity binding [40,48]. In UPF3B, both the RRM-L and the middle domain are involved in the interaction with UPF2 [44]. Among plant pathogenic fungi that contain *UPF3* homologous genes, only seven possess the RRM-L domain. Interestingly, the C-terminus of six of these plant pathogenic fungi also features additional RRM domains (Figure 4). This suggests that UPF3 in plant pathogenic fungi may interact with other factors via these additional RRM domains.

Above all, the majority of plant pathogenic fungi possess *UPF3* homologous genes, which include the key functional domain RRM-L. However, the degree of conservation is significantly lower in comparison to UPF1 and UPF2. UPF3 exhibits the greatest variation among the UPF genes, with some plant pathogenic fungi lacking homologous genes, while humans possess two. Furthermore, UPF3 have evolved RRM domains that demonstrate a stronger binding affinity compared to RRM-L in some plant pathogenic fungi, a finding that appears to be previously unreported.

### 3.5. Interaction of Nonsense-Mediated Decay Core Components in Plant Pathogenic Fungi

The NMD requires many proteins, UPF1, UPF2 and UPF3 constitute the basic mechanism. In the context of the transcript containing PTC, the eukaryotic release factors eRF1 and eRF2 interact with SMG1, SMG8, and SMG9, subsequently recruiting UPF1 via CBP80 to form the SURF (SMG1-UPF1-eRF1-eRF3) complex [22,49,50]. The SURF complex forms a bridge between the ribosome and UPF3 and UPF2 associated with EJC. Subsequently, after UPF1 is phosphorylated by SMG1, the NMD is initiated [45]. Furthermore, research has demonstrated that the interaction between UPF1 and UPF3B occurs independently of UPF2, as these two proteins can directly interact with each other without its involvement [51]. During the process of NMD, the interaction among UPF is essential. Therefore, we predict the interaction probabilities between UPF1, UPF2, and UPF3 in plant pathogenic fungi using AlphaFold3 (AF3).

AF3 demonstrates an unprecedented level of accuracy in predicting interactions between proteins. The DockQ probability judgment value, which assesses protein-protein interactions, can be derived from AF3. A DockQ value of 0.23 or higher indicates a potential for protein interaction [30,52]. In plant pathogenic fungi, the DockQ values for UPF range from a minimum of 0.244 to a maximum of 0.662 (Figure 5). This indicates a relatively high probability of interaction among UPF proteins across all plant pathogenic fungi. Among the UPF proteins, UPF1 and UPF2 exhibit the highest DockQ scores. However, the DockQ scores between UPF2 and UPF3 are lower when compared to those between UPF1 and UPF2 as well as between UPF1 and UPF3 (Figure 5). All the above results suggest that the interaction between UPFs in plant pathogenic fungi plays a crucial role in initiating the NMD.

## 4. Discussion

NMD has been widely studied in humans, especially in the direction of human diseases [53,54,55]. However, research on fungi, particularly plant pathogenic fungi, has been limited. This study clarifies the existence of core NMD genes in plant pathogenic fungi through the identification of homologous genes and the prediction of gene interactions, highlighting their similarities and differences with those found in animals.

NMD is widely present in plant pathogenic fungi. Both the *UPF* and members of the *EJC* genes are generally retained in these fungi. In NMD, the *UPF* genes and the EJC serve as important functional elements [32]. Their homologous genes have been identified in various plant pathogens. Importantly, the UPFsand EJCs, as core factors of NMD, exhibit high conservation in the critical functional protein domains. The CH domain of the UPF1, along with the MIF4G-3 and U1BD domains of the UPF2, as well as the RRM-L domain of UPF3, are well conserved in plant pathogenic fungi. This indicates that the functions of core NMD components in plant pathogenic fungi are largely similar to those of genes in humans. In plant pathogenic fungi, there is a significant likelihood that a NMD exists, which involves collaboration between the UPFs and EJCs.

The NMD exists with species specificity in plant pathogenic fungi. This study identifies three specific characteristics of the NMD in plant pathogenic fungi. The first characteristic is the uniqueness of the UPF1 in these fungi, which appears to be related to the absence of SMG1. In humans, the phosphorylation of UPF1 by SMG1 is crucial for initiating the NMD [46]. NMD is initiated only when UPF1 is phosphorylated. Notably, the SMG1 is typically absent in plant pathogenic fungi, mirroring the situation observed in *S. cerevisiae* [56]. In all plant pathogenic fungi, UPF1 lacks the SQ domain, and there is also no SMG1 responsible for phosphorylation. Therefore, in plant pathogens, two possibilities arise: either the UPF1 in these organisms does not require phosphorylation to function, or the phosphorylation in plant pathogenic fungi is of a different type that differs from that in humans. Further investigation is required to determine whether UPF1 can be phosphorylated by alternative genes in plant pathogenic fungi.

The second distinctive characteristic is the relatively poor conservation of the *UPF3* gene, which appears to be the last component formed during the evolution of the NMD. Among the three *UPF* genes, *UPF3* exhibits the greatest variability. As previously mentioned, in plant pathogenic fungi, there exists only one homologous gene for *UPF3*, and in some fungal species, there is no homolog of this gene at all. In humans, two *UPF3* homologous gene products are present; however, only one of them is functional, while the other is nearly non-participatory, indicating the evolutionary flexibility of *UPF3*.

The third distinctive characteristic is that homologous genes of the EJC component are present in all plant pathogenic fungi. Among all NMD components in plant pathogenic fungi, the eRF4A3 exhibits the highest homology to its human counterpart. This discovery is also the most significant finding in this study. NMD was initially discovered in *S. cerevisiae*; however, the limited presence of introns and the absence of EJC in *S. cerevisiae* restrict its reference significance for NMD in plant pathogenic fungi [10,57]. In addition, two components of the EJC, MAGOH and RBM8A, are present in plant pathogenic fungi but are absent in *S. cerevisiae*. Based on the aforementioned results, it can be inferred that a high likelihood exists for the presence of the NMD, which is dependent on the EJC, in plant pathogenic fungi. Plant pathogenic fungi exhibit a higher abundance of introns compared to *S. cerevisiae*. This suggests that a significant number of transcripts containing premature termination codons will be generated, and the requisite conditions for the NMD, which relies on the EJC-dependent pathway, are present. The existence of EJC homologous genes in plant pathogenic fungi is of considerable significance. If an EJC-dependent model exists in plant pathogenic fungi, they could regulate protein expression by altering the structure of the encoded proteins, thereby increasing protein diversity.

Based on the preceding analysis, we propose the following speculative perspectives. The genomes of plant pathogenic fungi demonstrate a higher density of introns, which consequently results in the production of a greater number of truncated proteins. These fungi have evolved mechanisms to degrade transcripts that contain PTC, a phenomenon corroborated by the identification of EJC within them. Therefore, we hypothesize that the NMD in plant pathogenic fungi exists in a transitional phase between EJC-independent and EJC-dependent. Although it is not identical to the NMD observed in humans, it bears closer resemblance to the human NMD than that found in *S. cerevisiae*.

Overall, this research conducts a comprehensive analysis of the conservativeness of the core elements of the NMD across ten significant plant pathogenic fungi worldwide. It offers new insights into the conservativeness and specificity of the NMD in these fungi. These novel perspectives on the NMD in plant pathogenic fungi lay the groundwork for future research aimed at exploring its influence on growth and development, as well as its role in pathogenic mechanisms.

## Figures and Tables

**Figure 1 jof-11-00404-f001:**
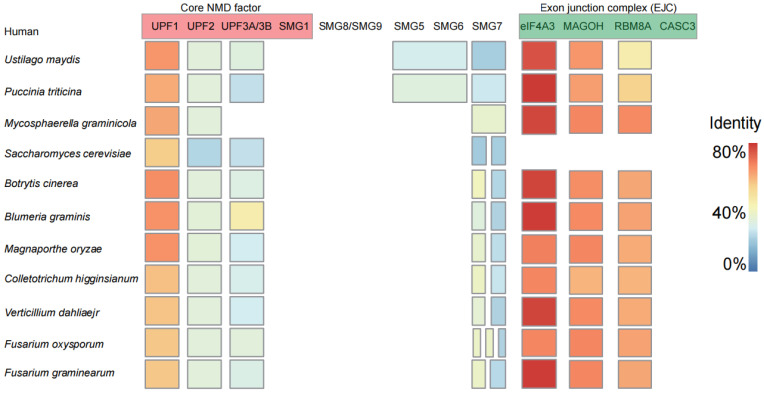
Identification of homologous genes of core components in the NMD pathway in plant pathogenic fungi. The red components denote the core NMD factors, which include *UPF1*, *UPF2*, and *UPF3A/B*. The green components represent the exon junction complex (EJC), comprising *eIF4A3*, *MAGOH*, *RBM8A*, and *CASC3*. High (orange to red) and low (yellow to blue) expression levels are represented by the sequence similarity of each gene. A gene represented by multiple squares indicates the presence of several homologous genes. Conversely, multiple genes sharing a single square signify that their homologous genes are identical. The absence of a square indicates that no homologous gene has been identified.

**Figure 2 jof-11-00404-f002:**
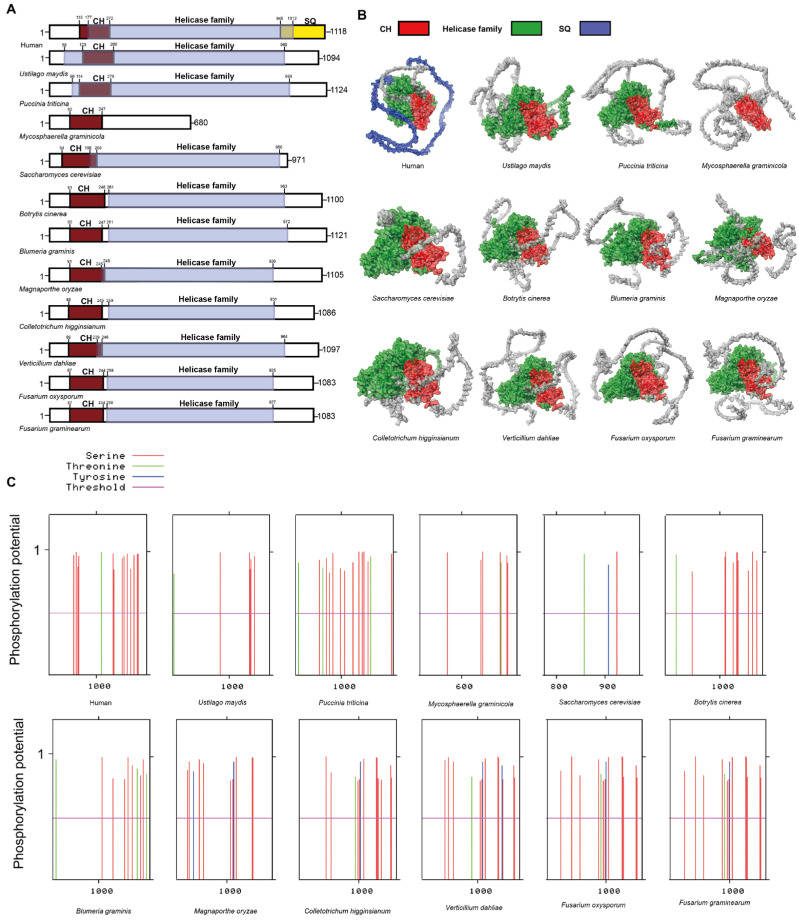
Functional domain analysis of UPF1. (**A**) Schematic diagram illustrating the domain architecture of UPF1. (**B**) Domain display in the UPF1. (**C**) The predicted phosphorylation sites in the UPF1.

**Figure 3 jof-11-00404-f003:**
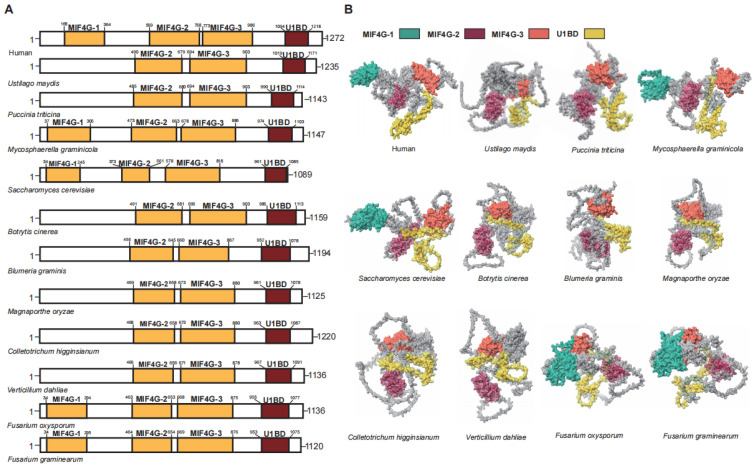
Functional domain analysis of UPF2. (**A**) Schematic diagram illustrating the domain architecture of UPF2. (**B**) Domain display in the UPF2.

**Figure 4 jof-11-00404-f004:**
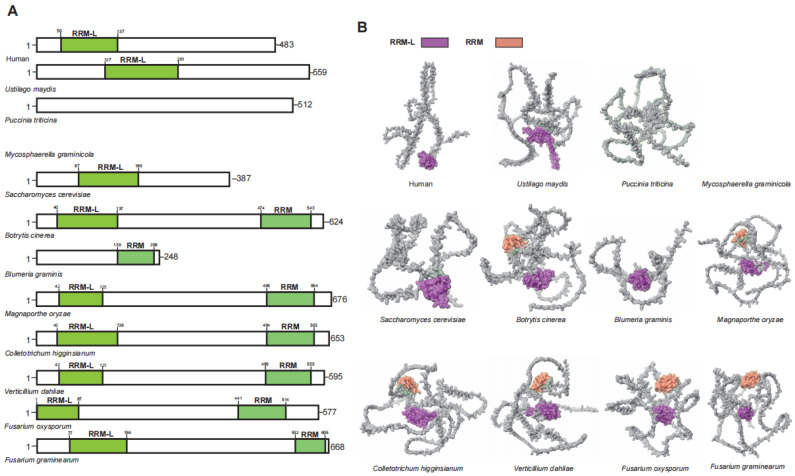
Functional domain analysis of UPF3. (**A**) Schematic diagram illustrating the domain architecture of UPF3. (**B**) Domain display in the UPF3.

**Figure 5 jof-11-00404-f005:**
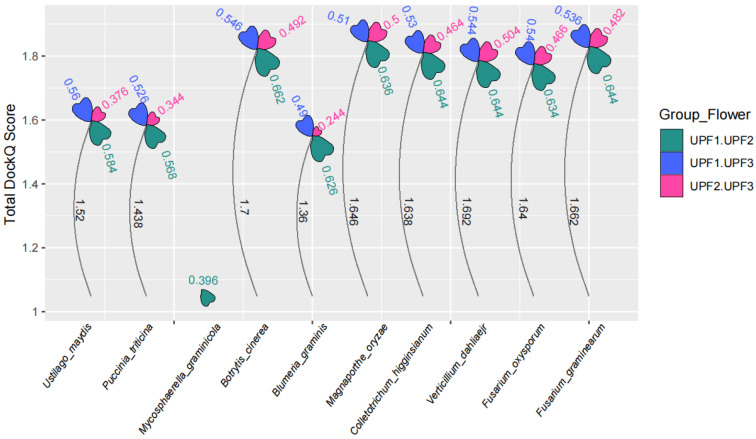
The interaction probabilities among the UPF1, UPF2, and UPF3 proteins in plant pathogenic fungi. The petal area represents the interaction scores, with different petals indicating the interaction scores between UPF1 and UPF2, UPF2 and UPF3, and UPF1 and UPF3. The height of the petals corresponds to the cumulative scores of interaction probabilities among the UPF proteins.

**Table 1 jof-11-00404-t001:** Research on the Mechanism of Nonsense-Mediated mRNA Decay (NMD).

Species	Year	Key Findings	Reference
*S. cerevisiae*	1979	NMD first discovered in eukaryotes.	[10]
Human	1979	The first discovery of thalassemia directly caused by NMD in humans.	[11]
*C. elegans*	1993	First demonstration of the importance of smg in NMD in nematodes.	[12]
*A. thaliana*	1998	The presence of NMD in plants was verified.	[14]
*Drosophilidae*	1999	NMD may affect the 3′ end processing of pre-mRNA.	[13]
*D. discoideum*	1999	NMD occurs in the nucleus in *D. discoideum*.	[15]
mice	2000	NMD is involved retinal function in mouse.	[16]
zebrafish	2009	NMD is involved in embryonic development in zebrafish.	[17]

## Data Availability

All raw data were sourced from publicly available websites and are described in the Materials and Methods section.

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
