# Peer review of "Up-Frameshift Factors from Phytopathogenic Fungi Play a Crucial Role in Nonsense-Mediated mRNA Decay"

_jof, 2025, doi:10.3390/jof11060404_

Round 1

Reviewer 1 Report

The manuscript by Ping Lu et al. presents an analysis of the nonsense-mediated mRNA decay phenomenon for phytopathogenic fungi based on bioinformatics studies. The work is based on the authors' hypotheses about the possible similarity of this phenomenon mechanism in pathogens and higher organisms.

My main criticism relates to the form of the results presentation.

The results presentation largely relates to the discussion of the results. In this regard, I would recommend that the authors combine the results and discussion sections. At the same time, this section should be carefully rethought, emphasizing the new data obtained by the authors, and more clearly supporting the conclusions made. In the presented form, the idea of the manuscript is lost in the abundance of references to other works, which reduces its merits and does not allow us to identify the significant achievements of the authors.

Minor comments

Figure 1, In my opinion, it would be more informative to replace it with a table in which the main points of the discovery of the mechanism of nonsense-mediated mRNA decay with their features in different organisms and the corresponding references should be noted.

233-244 – practically repeats the material presented in the introduction.

234 – contain premature termination codons (PTC). – the abbreviation was already introduced earlier, in the introduction section.

References (links) are not according to the rules, the Refs themselves are also not according to the rules.

Author Response

Reviewer1:

  1. I would recommend that the authors combine the results and discussion sections. At the same time, this section should be carefully rethought, emphasizing the new data obtained by the authors, and more clearly supporting the conclusions made. In the presented form, the idea of the manuscript is lost in the abundance of references to other works, which reduces its merits and does not allow us to identify the significant achievements of the authors.

Response: Many thanks! Both reviewers simultaneously pointed out the insufficient elaboration on the novel findings presented in this study. In response, we have emphasized the significance of the EJC-dependent mode in plant pathogens within the discussion section (line251-line256). We appreciate the reviewer's suggestion to merge the results and discussion sections, as it is a valuable recommendation. However, upon reviewing the journal's official guidelines, we have determined that both the results and discussion sections are essential components of a research manuscript (https://www.mdpi.com/journal/jof/instructions).

  1. Figure 1, In my opinion, it would be more informative to replace it with a table in which the main points of the discovery of the mechanism of nonsense-mediated mRNA decay with their features in different organisms and the corresponding references should be noted.

Response: Many thanks! We have replaced Figure 1 with Table 1 at line69. Due to the deletion of Figure 1, the numbering of the subsequent images is decremented accordingly (line94, line96, line99, line103, line118, line121, line122, line131, line139, line141, line143, line165, line170, line177, line199, line203, line207).

3.233-244 – practically repeats the material presented in the introduction..

Response: Many thanks! In this section, we have eliminated the redundant background that previously overlapped with the introduction. However, a small portion needs to be retained to introduce the necessity of undertaking this topic (line212-line216).

4.234 – contain premature termination codons (PTC). – the abbreviation was already introduced earlier, in the introduction section.

Response: Many thanks! In the previous question, the review suggested removing the redundant introductory section, and this issue happens to be within the deleted content.

4.References (links) are not according to the rules, the Refs themselves are also not according to the rules.

Response: Many thanks! We download the MDPI journal format from the Endenote official website and updated it.

Reviewer 2 Report

Comments: This manuscript presents a comprehensive and well-organized study on the identification and characterization of nonsense-mediated mRNA decay (NMD) core factors in ten economically significant phytopathogenic fungi. The manuscript offers a valuable foundation for future investigations into the roles of NMD in fungal development and pathogenicity.

However, minor revisions are necessary to improve clarity, correct minor typographical errors, and strengthen the readability of the text.

Language/Grammar:

  • Please revise the manuscript for minor grammatical issues and awkward phrasing. A few examples include:
      • "a gene coded the Methyl-CpG binding protein 2" → "a gene encoding the Methyl-CpG binding protein 2."
      • "functions relate to epigenetic silencing" → "functions related to epigenetic silencing."
      • "Nonsense-mediated decay mechanism" should often be "nonsense-mediated decay (NMD)" without repeating "mechanism" unnecessarily.
      • In places like "The NMD is widely present in plant pathogenic fungi", consider removing the definite article "The" unless referring to a specific instance.
  • Minor rephrasing for smoothness could be helpful, e.g., "Surprisingly, homologous genes of UPF3 are absent in individual plant pathogenic fungi" might better read "Interestingly, some plant pathogenic fungi lack homologous genes of UPF3."

Formatting/Consistency:

    • Ensure consistent use of gene/protein names (e.g., SMG1, UPF1, eIF4A3) in italics or standard format throughout, according to journal style.
    • Abbreviations like NMD and EJC are well-defined, but after initial definition, be consistent in their use.
    • Website URLs should be carefully checked and properly formatted without extra characters.

Introduction section:

  • In the introduction, it would be helpful to briefly explain what an exon junction complex (EJC) is when first mentioned, for broader accessibility.
  • Some terms like "SMG genes" are introduced without immediately explaining the acronym (Suppressor with Morphological effect on Genitalia). Please ensure such terms are properly introduced.
    • Consider adding a sentence to highlight the biological implications of the presence of EJC components in phytopathogenic fungi, as this is a notable finding.

Discussion:

    • In the discussion of EJC-dependent versus EJC-independent pathways, it would be helpful to briefly hypothesize the biological significance of EJC presence in plant pathogenic fungi compared to its absence in S. cerevisiae (for instance, linking to pathogenicity or lifecycle complexity).
    • Some sentences in the Discussion could be improved for clarity and readability. For example, phrases like "relatively significant UPF genes" could be more precisely worded.

Methodology:

    • In Section 4.2, specify if an E-value cutoff or any threshold was applied during the BLASTp alignment.
    • In Section 4.4, briefly explain the interpretation of DockQ scores for readers who may not be familiar with this evaluation metric.

Comments: This manuscript presents a comprehensive and well-organized study on the identification and characterization of nonsense-mediated mRNA decay (NMD) core factors in ten economically significant phytopathogenic fungi. The manuscript offers a valuable foundation for future investigations into the roles of NMD in fungal development and pathogenicity.

However, minor revisions are necessary to improve clarity, correct minor typographical errors, and strengthen the readability of the text.

Language/Grammar:

  • Please revise the manuscript for minor grammatical issues and awkward phrasing. A few examples include:
      • "a gene coded the Methyl-CpG binding protein 2" → "a gene encoding the Methyl-CpG binding protein 2."
      • "functions relate to epigenetic silencing" → "functions related to epigenetic silencing."
      • "Nonsense-mediated decay mechanism" should often be "nonsense-mediated decay (NMD)" without repeating "mechanism" unnecessarily.
      • In places like "The NMD is widely present in plant pathogenic fungi", consider removing the definite article "The" unless referring to a specific instance.
  • Minor rephrasing for smoothness could be helpful, e.g., "Surprisingly, homologous genes of UPF3 are absent in individual plant pathogenic fungi" might better read "Interestingly, some plant pathogenic fungi lack homologous genes of UPF3."

Formatting/Consistency:

    • Ensure consistent use of gene/protein names (e.g., SMG1, UPF1, eIF4A3) in italics or standard format throughout, according to journal style.
    • Abbreviations like NMD and EJC are well-defined, but after initial definition, be consistent in their use.
    • Website URLs should be carefully checked and properly formatted without extra characters.

Introduction section:

  • In the introduction, it would be helpful to briefly explain what an exon junction complex (EJC) is when first mentioned, for broader accessibility.
  • Some terms like "SMG genes" are introduced without immediately explaining the acronym (Suppressor with Morphological effect on Genitalia). Please ensure such terms are properly introduced.
    • Consider adding a sentence to highlight the biological implications of the presence of EJC components in phytopathogenic fungi, as this is a notable finding.

Discussion:

    • In the discussion of EJC-dependent versus EJC-independent pathways, it would be helpful to briefly hypothesize the biological significance of EJC presence in plant pathogenic fungi compared to its absence in S. cerevisiae (for instance, linking to pathogenicity or lifecycle complexity).
    • Some sentences in the Discussion could be improved for clarity and readability. For example, phrases like "relatively significant UPF genes" could be more precisely worded.

Methodology:

    • In Section 4.2, specify if an E-value cutoff or any threshold was applied during the BLASTp alignment.
    • In Section 4.4, briefly explain the interpretation of DockQ scores for readers who may not be familiar with this evaluation metric.

Author Response

Reviewer2:

  1. Please revise the manuscript for minor grammatical issues and awkward phrasing. A few examples include:

"a gene coded the Methyl-CpG binding protein 2" → "a gene encoding the Methyl-CpG binding protein 2."

"functions relate to epigenetic silencing" → "functions related to epigenetic silencing."

"Nonsense-mediated decay mechanism" should often be "nonsense-mediated decay (NMD)" without repeating "mechanism" unnecessarily.

In places like "The NMD is widely present in plant pathogenic fungi", consider removing the definite article "The" unless referring to a specific instance.

Response: Many thanks! We have revised the following grammatical content. We have changed the "a gene coded the Methyl-CpG binding protein 2" to "a gene encoding the Methyl-CpG binding protein 2." at line39.

We have confirmed that the content in the original text is correct. The sentence in the text is "functions related to epigenetic silencing," not "functions relate to epigenetic silencing.". at line39.

We have changed the "Nonsense-mediated decay mechanism" to "nonsense-mediated decay". According to the suggestion in Article 4, we will convert all instances of Nonsense-mediated mRNA decay to its abbreviation (NMD), except for the first occurrence at line3, line13, line15, line16, line19, line26, line27, line51, line55, line60, line75, line104, line113, line135, line157, line184, line192, line217, line226, line227, line235, line237, line265, line270.

We have changed the "The NMD is widely present in plant pathogenic fungi" to "NMD is widely present in plant pathogenic fungi" at line217.

  1. Minor rephrasing for smoothness could be helpful, e.g., "Surprisingly, homologous genes of UPF3 are absent in individual plant pathogenic fungi" might better read "Interestingly, some plant pathogenic fungi lack homologous genes of UPF3."

Response: Many thanks! We have changed the "Surprisingly, homologous genes of UPF3 are absent in individual plant pathogenic fungi" to "Interestingly, some plant pathogenic fungi lack homologous genes of UPF3." at line234-line235.

  1. Ensure consistent use of gene/protein names (e.g., SMG1, UPF1, eIF4A3) in italics or standard format throughout, according to journal style.

Response: Many thanks! We ensured onsistent use of gene/protein names in italics or standard format throughout at line20, line22, line29, line98, line99, line101, line105, line106, line158, line166, line175, line179, line192, line198, line200, line201, line207, line210, line217, line218, line220, line221, line228, line235, line279.

  1. Abbreviations like NMD and EJC are well-defined, but after initial definition, be consistent in their use.

Response: Many thanks! After the initial definitions of NMD and EJC, we consistently employ these abbreviations throughout the main text at line3, line13, line15, line16, line19, line26, line27, line51, line55, line60, line67, line71, line72, line75, line104, line113, line135, line157, line184, line188, line192, line217, line218, line220, line226, line227, line235, line237, line239, line240, line265, line270. The full terms are utilized exclusively in figure captions, abstracts, and titles, as these sections must ensure that readers can comprehend their meanings even when read independently.

  1. Website URLs should be carefully checked and properly formatted without extra characters.

Response: Many thanks! We have verified all the URLs and ensured that the websites can be accessed without extra characters.

  1. In the introduction, it would be helpful to briefly explain what an exon junction complex (EJC) is when first mentioned, for broader accessibility.

Response: Many thanks! We briefly explain what an exon junction complex (EJC) in the introduction at line71-line74.

  1. Some terms like "SMG genes" are introduced without immediately explaining the acronym (Suppressor with Morphological effect on Genitalia). Please ensure such terms are properly introduced.

Response: Many thanks! We ensured that the full name of the SMG1(suppressors with morphological effects on genitalia proteins) gene was introduced upon its first time in the abstract at line21-line22.

  1. Consider adding a sentence to highlight the biological implications of the presence of EJC components in phytopathogenic fungi, as this is a notable finding.

Response: Many thanks! We emphasized the biological significance of the EJC components in phytopathogenic fungi in the discussion at line251-line256.

  1. In the discussion of EJC-dependent versus EJC-independent pathways, it would be helpful to briefly hypothesize the biological significance of EJC presence in plant pathogenic fungi compared to its absence in S. cerevisiae(for instance, linking to pathogenicity or lifecycle complexity).

Response: Many thanks! In light of the repetitiveness with the previous suggestion we have consolidated the presence of EJC in plant pathogenic fungi along with the significance of its existence into a single, cohesive discussion. This approach enhances clarity and allows for a more comprehensive examination of the topic at line251-line256.

  1. Some sentences in the Discussion could be improved for clarity and readability. For example, phrases like "relatively significant UPF genes" could be more precisely worded.

Response: Many thanks! We have changed the "Importantly, the relatively significant UPF genes and the exon junction complex (EJC) exhibit a high degree of conservation in their functional protein domains." to "Importantly, UPF genes and EJC, as core factors of NMD, exhibit high conservation in the critical functional protein domains." at line220-line221.

  1. In Section 4.2, specify if an E-value cutoff or any threshold was applied during the BLASTp alignment.

Response: Many thanks! We already added the E-value of blastp for genes identified as homologous at line272-line273.

  1. In Section 4.4, briefly explain the interpretation of DockQ scores for readers who may not be familiar with this evaluation metric.

Response: Many thanks! We have explained the interpretation and calculation method of DockQ scores at line281-line284.

Round 2

Reviewer 1 Report

I thank the authors for their responses to my comments and corrections. However, the main criticism I made in the first review, in my opinion, was not sufficiently taken into account. I still believe that the results section should clearly describe the novelty of the data obtained by the authors, and the discussion should be based on them. The presented version contains almost no necessary changes.

no

Author Response

  1. The results section should clearly describe the novelty of the data obtained by the authors, and the discussion should be based on them.

Response: We greatly appreciate the reviewer's insightful feedback and place significant importance on the suggestions provided. We have incorporated additional content into the results section to provide a clearer description of the novelty of the data obtained. This includes highlighting the significance of identifying EJC in plant pathogenic fungi (line101-line105), as well as emphasizing the specificity of UPF1 (line133-line137), UPF2 (line163-line165) and UPF3 (line191-line195).To enhance the prominence of the key findings in this study, we have restructured the discussion section. The new findings are categorized into three distinct aspects, each of which has been discussed individually to provide clarity and depth (line238-line274).

Round 3

Reviewer 1 Report

No

The authors have improved the results and discussion section. Now the material looks more structured. I would recommend emphasizing the novelty of the research more. But I leave this remark to the editor-in-chief.

Author Response

Reviewer1:

  1. The authors have improved the results and discussion section. Now the material looks more structured. I would recommend emphasizing the novelty of the research more. But I leave this remark to the editor-in-chief.

Response: Many thanks!We have additionally included content to emphasize the novelty of this article. Based on the experimental results, this study proposes that the NMD in plant pathogenic fungi is undergoing an evolutionary transition from EJC-independent to EJC-dependent, and has already acquired the ability to degrade transcripts containing PTCs. Although a few components exhibit some differences compared to those in humans, overall, the mechanism is more similar to that in humans.